# Repeatability of flicker modulation sensitivity measured using the Flicker-*Plus* test

**Aiman Hafeez[1]\*, Alison Binns[1], Sajni Bohra[1], Irene Ctori[2], John L. Barbur[1]**

**1** School of Health and Medical Sciences, Department of Optometry and Visual Science, City, St. George's University of London, London, United Kingdom, **2** School of Health, Medicine, and Life Sciences, University of Hertfordshire, Hatfield, London, United Kingdom

\* aiman.hafeez@citystgeorges.ac.uk

## Abstract

### Purpose

To evaluate the test-retest variability of the Flicker-*Plus* test for each of the two protocols measuring rod and cone-enhanced flicker modulation thresholds (FMT) in healthy individuals. A secondary aim was to evaluate the within-subject variability in repeated measurements.

### Methods

Thirty healthy participants aged 19–71 years were examined. None had any history or signs of ocular disease. Monocular FMT were measured at the fovea (0°) and at an eccentricity of 5° in each quadrant, twice by the same investigator under identical conditions within a 2-week period under stimulus conditions that favoured either rods or cones to evaluate the between session repeatability. To assess the within-subject variability, binocular measurements for cone and rod-enhanced FMT were carried out on 15 different occasions over a period of 3-weeks in three of the participants. Coefficient of Repeatability (CoR) was calculated for inter-session repeatability and Bland Altman plots were created for graphical representation. Inter-class correlation coefficient (ICC) with 95% confidence intervals were estimated.

### Results

Bland and Altman analysis shows that the mean bias is greater than zero in all 5 testing locations for both rod and cone-enhanced FMTs, suggesting that the threshold at the second visit tended to be lower than at the first, however the difference between visits was not statistically significant for any test condition (paired t-test, $p < 0.05$). In a sub analysis for those CoR was found to be higher in those aged <45years, compared to those aged ≥45 years. The correlation and agreement between the two

**Data availability statement:** The dataset file is available from Figshare Repository at: https://doi.org/10.25383/city.29891168.

**Funding:** The author(s) received no specific funding for this work.

measurements estimated by ICC analysis shows good (0.75–0.9) to excellent (>0.9) test-retest reliability of Flicker-*Plus* test for all measures.

## Conclusion

The findings show good to excellent test-retest repeatability for the Flicker-*Plus* test. This was the case at all locations (foveal and peripheral), under both cone and rod-enhanced conditions. There was no evidence of significant learning effects.

---

## Introduction

Flicker sensitivity can be measured by assessing the smallest flicker modulation a subject just needs to see flicker when viewing a temporally modulated stimulus. Most tests systematically adjust the light modulation amplitude at a fixed temporal frequency without changing the mean light level of the stimulus [1]. A staircase procedure is normally employed to measure a threshold modulation amplitude that corresponds to a predetermined probability of a correct response. The reciprocal of this threshold modulation signal is a measure of flicker sensitivity [2]. The retina serves as one of the most highly active metabolic tissues in the body, necessitating a continuous supply of oxygen by the retinal and choroidal vessels [3]. Studies investigating enzyme activity and oxygen utilising micro-electrodes have shown that the layer of retina containing the outer segments of photoreceptors is the most metabolically active [4,5]. Flickering light stimuli presented at temporal frequencies above 10 Hz have been shown to increase neural activity, blood flow, and haemoglobin oxygen saturation in retinal veins [6,7]. As a result, retinal oxygen metabolism increases in response to the physiological challenge of high frequency temporal stimulation [8]. It follows that loss of rapid flicker sensitivity may provide useful information for early detection of retinal pathologies such as age-related macular degeneration (AMD) [9–11], glaucoma [12,13], and diabetic retinopathy [14], where hypoxia is a contributing factor in disease progression. Rapid flicker thresholds have high sensitivity and reproducibility and can be useful in assessing the integrity of photoreceptors [15] and any changes in visual function that may arise early in retinal diseases such as AMD [8]. Similarly in glaucoma, the measurement of increased optic nerve head blood flow caused by flicker driven neural activity can be useful for understanding the pathogenesis of glaucomatous optic nerve damage [13]. It is reported that flicker modulation sensitivity is substantially altered in patients with ocular hypertension (OHT) and in early open angle glaucoma [13,16–18]. Despite these potential clinical applications and interest in flicker sensitivity, the number of published studies on test-retest repeatability of flicker thresholds are few with only small number of testing methods investigated [9,19,20]. A knowledge of expected inter-subject and within-subject variability in flicker modulation thresholds (FMT) as well as normal limits of flicker thresholds as a function of age would be of great value in the detection of abnormal responses and for the management of patients with retinal disease.

The Flicker-*Plus* test measures temporal contrast sensitivity under photopic (cone-enhanced) and mesopic (rod-enhanced) conditions. FMTs are measured in central vision and at four discrete diagonal, peripheral locations (5° in each quadrant). The two protocols isolate the function of the rods and cones by appropriate selection of test parameters including retinal illuminance, spectral composition, and spatio-temporal characteristics, that favour either rods or cones. In addition to measuring thresholds at five discrete locations in the visual field, the test requires minimal dark adaptation time.

The primary aim of this study was to assess the variability in test-retest repeats of the AVOT Flicker-*Plus* test for each of the two protocols measurements of rod- and cone-enhanced FMT in healthy individuals. A secondary aim was to evaluate the within subject variability in repeated measurements.

## Methods

### Participants

Thirty healthy participants were recruited through adverts placed in cafes, supermarkets, libraries, GP surgeries, the City University eye clinic, through social media and advertising platforms and from City, St. George's University of London students and staff members between 01st November 2023–31st March 2024. To be eligible for inclusion, participants were required to be ≥ 18 years of age, with best corrected visual acuity within the normal range in their test eye for the corresponding age [21], refractive error ≤ ± 6 dioptres and/or ± 3 dioptres of astigmatism with open-anterior chamber angle (grade 3 or 4 with Van-Herick technique) [22]. The status of the ocular media was also checked. Fundus imaging and disc examination was carried out using a slit lamp. Only those subjects with ≤ 2 on all Lens opacities and classification system III (LOCS III) criteria [23], intraocular pressure ≤ 21 mmHg (on non-contact tonometry), and no history of major systematic disease or medication known to affect retinal structure and function were recruited for the study.

### Ethical consideration and consent

This study was approved by the Senate Research Ethics Committee of School of Health and Medical Sciences City, St. George's University of London (Ethics reference number: ETH2122−1185) and the study fully complied with the standards stated in the Declaration of Helsinki. All participants were given an information leaflet 24 hours before their first visit. The investigator talked through the information leaflet with the participant at the start of the visit and encouraged each participant to ask any questions they may have before consent was obtained. A written informed consent form was received from each subject before any measurements were carried out.

### Testing procedure

Data collection took place at City, St. George's University of London. Data were collected by two investigators, AH and SB. In order to evaluate the between session repeatability of monocular FMT, the Flicker-*Plus* test was carried out twice with each participant by the same investigator under identical conditions within a 2-week period. To carry out an in-depth evaluation of the within subject variability of the test, measurements for cone and rod-enhanced FMT were carried out on 15 different occasions over a period of 3-weeks in three of the participants.

Screening data were collected at the start of the first visit to confirm that eligibility conditions were fulfilled. The evaluation included a patient's medical and ocular history, brief refraction, monocular logMAR visual acuity (ETDRS chart), K-readings, axial length (ALADDIN (Version 1.6.2, Topcon, Tokyo, Japan)), central corneal thickness and non-contact IOP measurement (Topcon TRK-1P (version 1.46.14) Auto Kerato-Refracto-Tonometer instrument (Topcon, Tokyo, Japan)), non-dilated fundus imaging and macular OCT scan (Topcon Maestro), visual field assessment (SITA standard 24−2, Humphrey Field Analyser), non-dilated slit lamp examination of anterior and posterior segment along with lens clarity assessment and evaluated according to the LOCS III. Both eyes were evaluated to determine eligibility and, in the event that

both eyes met the inclusion criteria, the eye with better visual acuity was included for further assessment or, in the case of equal visual acuity, a randomisation schedule was used to select the test eye.

To measure the flicker thresholds, the Flicker-*Plus* test was run on a desktop computer. The standard protocols for mesopic (rod-enhanced) and photopic (cone-enhanced) stimulus conditions were used which vary in stimulus presentation duration (longer for rod-enhanced stimuli), temporal modulation frequency (lower for rod-enhanced stimuli), background luminance and spectral composition. The flickering disc stimulus was presented randomly at one of five possible locations in the visual field on a fully calibrated high-resolution monitor (Eizo CS2420). The display supports 10-bit dynamic range with a spatial resolution of 1920 x 1200 pixels at a frame rate of 60 Hz. The disc stimulus luminance was modulated sinusoidally at 5 Hz for the rod-enhanced condition and at 15 Hz (with square-wave modulation) for the cone-enhanced condition. The mean luminance of the stimulus was equal to that of the adjacent background (0.5 cd/m$^2$ for mesopic and 22 cd/m$^2$ for photopic protocol). The photopic central stimulus was of 0.5° diameter (30 minute of arc) and the parafoveal stimuli subtended a visual angle of 1° (60 minute of arc). Photopic stimuli were presented for 344ms. Under the mesopic protocol, the stimulus size for central and parafoveal locations was 0.75° (45 minute of arc) and 1.5° (90 minute of arc), respectively, and stimuli were presented for 600 ms. A curtain was used to isolate the stimulus display from a second display used by the examiner. A small, 3W LED lamp placed on the examiner's desk was used to provide mesopic ambient lighting. The examiner's display background also provided some ambient illumination. Within the subject's area, almost all background lighting was derived from the uniform background of the stimulus display. The spectral composition of the light employed for the two stimulus conditions was selected to generate a scotopic to photopic (S/P) luminance ratio of 0.9 for cone-enhanced conditions and 9 for rod-enhanced conditions [24]. Each participant was seated at 1m from the stimulus monitor, with the hood placed in front to position the forehead and chin at a fixed distance. The location of the stimulus was selected randomly. Before each stimulus, the central fixation target (consisting of an outline square and cross) flashed briefly to inform the subject of the next stimulus which followed with a delay varying from 600 to 1000ms. The participants were instructed to look at the central fixation target and to keep fixation on the centre of the

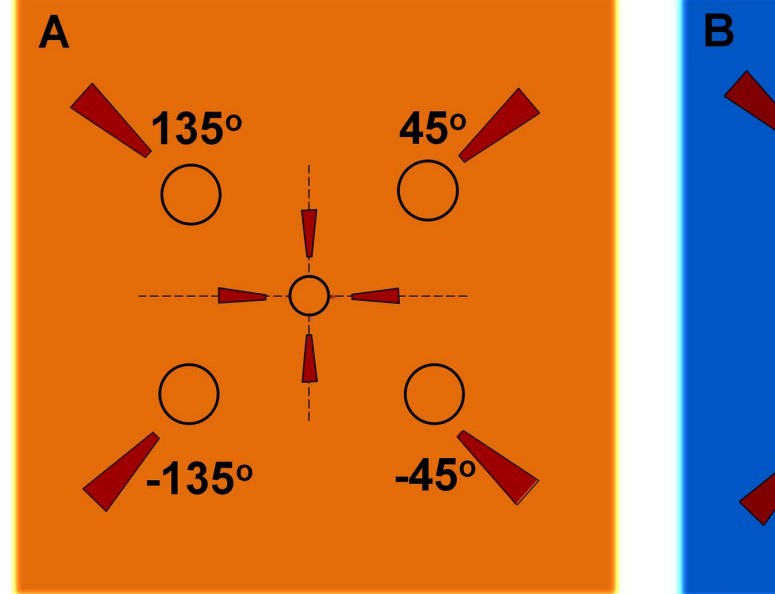
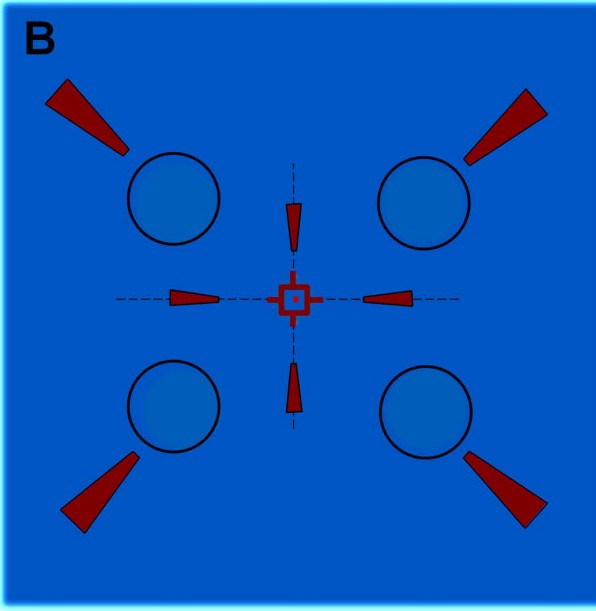

**Fig 1. Schematic presentation of cone-enhanced (A) and rod-enhanced (B) testing panel.** The circles with numbers in (A) indicate the locations of stimulus in four peripheral locations. The central fixation square is located at the location of foveal stimulus presentation point (B).

screen as indicated by pointing arrows (Fig 1). The display background consisted solely of mid-to-long wavelength light (CIE $x$ = 0.413, y = 0.507) to reduce variations in short-wavelength light absorption caused by the crystalline lens and macular pigment. Monocular thresholds were measured with rod and cone enhanced stimuli at the fovea (0 degrees) and at four parafoveal locations, diagonally displaced 5° away from fixation into each quadrant. The flicker thresholds were measured at each of the five locations randomly using five interleaved 2-down 1-up adaptive staircases with 10 reversals. The threshold was based on the average of the last six reversals. Participants were provided with a numeric keypad with 5 keys that matched spatially the stimulus positions on the screen. A separate key (green) placed well above the others was used to indicate when the subject was completely unaware of the stimulus position in the visual field. This response button, when pressed, instructs the program to allocate randomly the response to any of the five possible stimulus locations. The task was explained to each participant and the instructions were to indicate the location of the stimulus presented on the screen by pressing the corresponding keypad button. The 'learning mode' (which employed suprathreshold stimuli) was used initially to familiarise each participant with the procedure before the full test was administered. A brief fixation stimulus made up of a dark square outline and a cross preceded each stimulus presentation. This brief flash attracted the subject's fixation to the centre of the screen in anticipation for the next stimulus. The participant was required to keep their gaze fixed on this marker throughout the test in order to see the peripherally displayed stimuli at 5° eccentricity. Each subject, before being tested in the rod-enhanced condition, had to adapt to the background field for ~2 minutes while wearing spectrally calibrated neutral density filters (Oakley Half Jacket 2.0 –Black Iridium, USA) selected to reduce approximately 10-fold the light flux entering the eye, which were worn throughout the rod-enhanced testing phase. The use of these filters made it possible to achieve a luminance of 0.5 cd/m² without compromising the expected visual display performance at very low light levels. Preliminary experiments showed that extending the background adaptation period for rod-enhanced condition from 1 to 16 minutes had no significant effect on the thresholds [25]. Each protocol took ~8 minutes to complete. The same testing procedure was repeated on follow-up visit(s). The FMTs are expressed on a linear scale as a percentage of the background luminance.

## Statistical analysis

Data were imported into Microsoft Excel 2017 and analysed using IBM SPSS® Statistical version 29 for Windows (SPSS, Chicago, USA). The Shapiro-Wilk test was used to evaluate the normality of data distribution. The pre-set level of significance was $\alpha$ = 0.05, with $p < 0.05$ indicating non-normal distributions. Normally distributed data were described by the mean and standard deviation (SD), whilst non normally distributed data were represented by the median and IQR. Flicker thresholds from one eye were used for inter-session repeatability analysis. The coefficient of repeatability (CoR) was calculated by multiplying the standard deviation (SD) of the differences between the FMTs recorded at both visits by 1.96 [26]. Bland Altman plots were created to graphically represent the repeatability of FMTs [26], demonstrating the agreement between the two measurements obtained on the same subject, under identical conditions within two weeks. The y-axis displays the difference between two measurements, while the x-axis represents the average of the repeated measurements. The middle horizontal solid line on the graph represents the average difference in measurements, also referred to as bias. The 95% upper and lower limits of agreement (LoA) represented by dashed lines (mean bias ± 1.96 x SD) were calculated based on the standard deviation of the inter-session differences between the FMT measurements. The effect of healthy ageing on FMT becomes more prominent after 45 years of age, with increasing age resulting in elevated thresholds, especially with respect to rod mediated responses [25], therefore an additional sub analysis was carried out to compare the CoR of FMT measured in individuals aged <45 years, and those of individuals aged 45 years or older. In order to compare CoR in proportional terms, it was also expressed as a percentage of the mean FMT for younger and older subgroups (% CoR). Inter-class correlation coefficients (ICC) with 95% confidence intervals were estimated based on a mean-rating (k = 2), absolute-agreement, and 2-way mixed effect model [27,28]. A paired sample t-test was conducted prior to Bland Altman plot construction to check if there was statistically significant difference between the mean cone and

rod FMTs for visit 1 and visit 2, which might indicate a learning effect. An independent sample t-test was conducted to compare FMTs between younger (<45 years) and older age groups (≥45 years).

For intra subject variability, standard deviation (SD) and coefficient of variation (CoV; CoV = SD/mean) of all measurements was calculated. The CoV is a measure of the dispersion of data relative to the mean and provides a measure of the degree of variation detected between repeated measurements that were taken from the same person on multiple occasions [29].

## Results

Thirty healthy participants (24 females and 6 males) aged 19–71 years (mean 44 years ±15SD were tested using Flicker-*Plus* test (Table 1). The average time between visits was 8 days (±2 SD). All data were normally distributed, hence parametric tests were used in the analysis presented below.

Mean values for FMT at all locations, in rod and cone-enhanced conditions are given in Tables 2 and 3, respectively. The CoR tended to be higher for rod-enhanced than cone enhanced conditions, and for foveal compared to parafoveal locations. When CoR was expressed as a proportion of the mean FMT to facilitate comparison (% CoR), the cone- and rod-enhanced stimuli showed broadly comparable repeatability (mean % CoR across locations: 44.1% vs. 41.8% for cones and rods, respectively). Additionally, foveal locations exhibited notably higher % CoR than parafoveal locations, especially for cones, indicating poorer repeatability at the fovea.

Between session repeatability data are represented graphically in the Bland Altman plots in Figs 2A–2E and 3A–3E. Thresholds are presented as percentages. The mean bias is greater than zero in all 5 testing locations for both rod- and cone-enhanced FMT, suggesting that the threshold at the second visit tended to be lower than at the first. However, paired sample t-tests showed the mean difference between visits to be statistically non-significant ($p < 0.05$) for all five-testing locations in cone and rod mediated responses, suggesting that there was no statistically significant learning effect. The 95% LoAs were wider for rod-mediated thresholds in all 5 testing locations, and foveal rod and cone thresholds had wider

**Table 1. Total number of participants recruited and participants per decade with mean age (M) and standard deviation (SD).**

|  | Number of Participants (n) | Mean age (M±SD) |
|---|---|---|
| Total | 30 | 44 ± 15.00 |
| 18 - 29 | 6 | 25 ± 4.32 |
| 30 - 39 | 6 | 33 ± 3.38 |
| 40 - 49 | 6 | 44 ± 1.21 |
| 50 - 59 | 6 | 53 ± 2.58 |
| 60 - 69 | 5 | 66 ± 1.30 |
| ≥ 70 | 1 | 71 ± 0.00 |

**Table 2. Inter visit repeatability of cone-enhanced FMT at all testing locations.**

| Cone-enhanced Testing location | Mean Thresholds of Visit 1 & Visit 2 | Mean difference between visits | CoR | % CoR | Lower Limit of Agreement | Upper Limit of Agreement |
|---|---|---|---|---|---|---|
| Inferonasal (−135) | 5.40 | 0.42 | 1.74 | 32.2 | −1.32 | 2.16 |
| Inferotemporal (−45) | 4.96 | 0.46 | 2.17 | 43.8 | −1.71 | 2.63 |
| Foveal (0) | 5.08 | 0.57 | 3.25 | 63.9 | −2.67 | 3.82 |
| Superotemporal (45) | 4.80 | 0.54 | 1.81 | 37.7 | −1.27 | 2.34 |
| Superonasal (135) | 4.68 | 0.49 | 2.00 | 42.7 | −1.50 | 2.49 |

**Table 3. Inter visit repeatability of rod-enhanced FMT at all testing locations.**

| Rod-enhanced Testing location | Mean Thresholds of Visit 1 & Visit 2 | Mean difference between visits | CoR | % CoR | Lower Limit of Agreement | Upper Limit of Agreement |
|---|---|---|---|---|---|---|
| Inferonasal (−135) | 6.85 | 0.56 | 2.37 | 34.6 | −1.81 | 2.92 |
| Inferotemporal (−45) | 6.69 | 0.39 | 2.98 | 44.5 | −2.59 | 3.36 |
| Foveal (0) | 7.69 | 1.36 | 3.34 | 43.4 | −1.98 | 4.70 |
| Superotemporal (45) | 6.24 | 0.07 | 3.30 | 52.9 | −3.23 | 3.36 |
| Superonasal (135) | 6.55 | 0.20 | 2.20 | 33.6 | −1.99 | 2.19 |

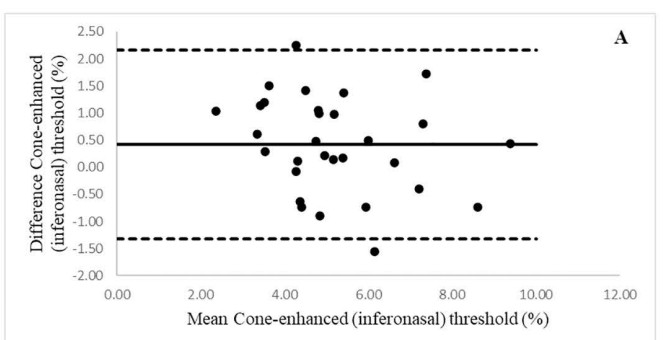
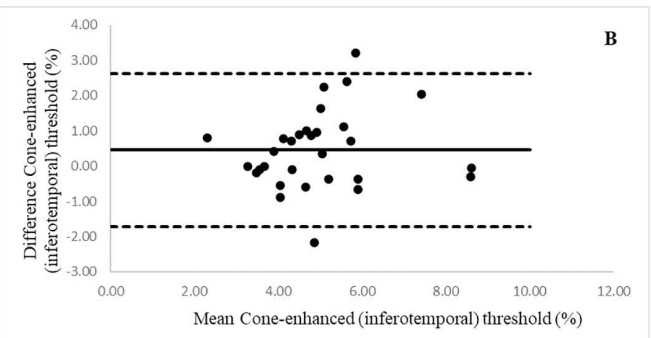
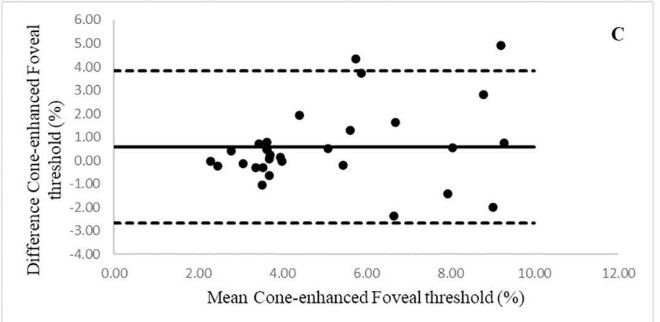
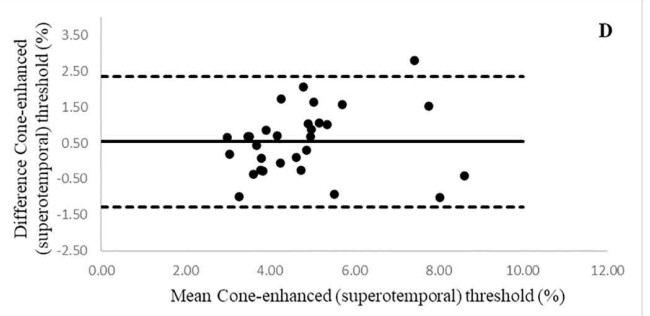
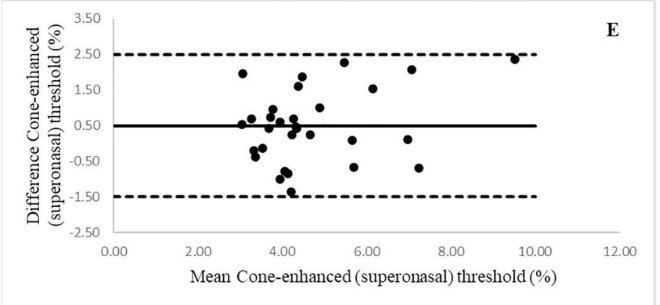

**Fig 2. Bland Altman plots to show between session repeatability.** (A) inter-visit agreement of inferonasal (−135) cone-enhanced FMT, (B) inter-visit agreement of inferotemporal (−45) cone-enhanced FMT, (C) inter-visit agreement of foveal (0) cone-enhanced FMT, (D) inter-visit agreement of super-otemporal (45) cone-enhanced FMT, (E) inter-visit agreement of superonasal (135) cone-enhanced FMT. Middle horizontal solid line represents mean difference between visits cone-enhanced. Upper and lower LoA (mean±1.96) represented by dashed line.

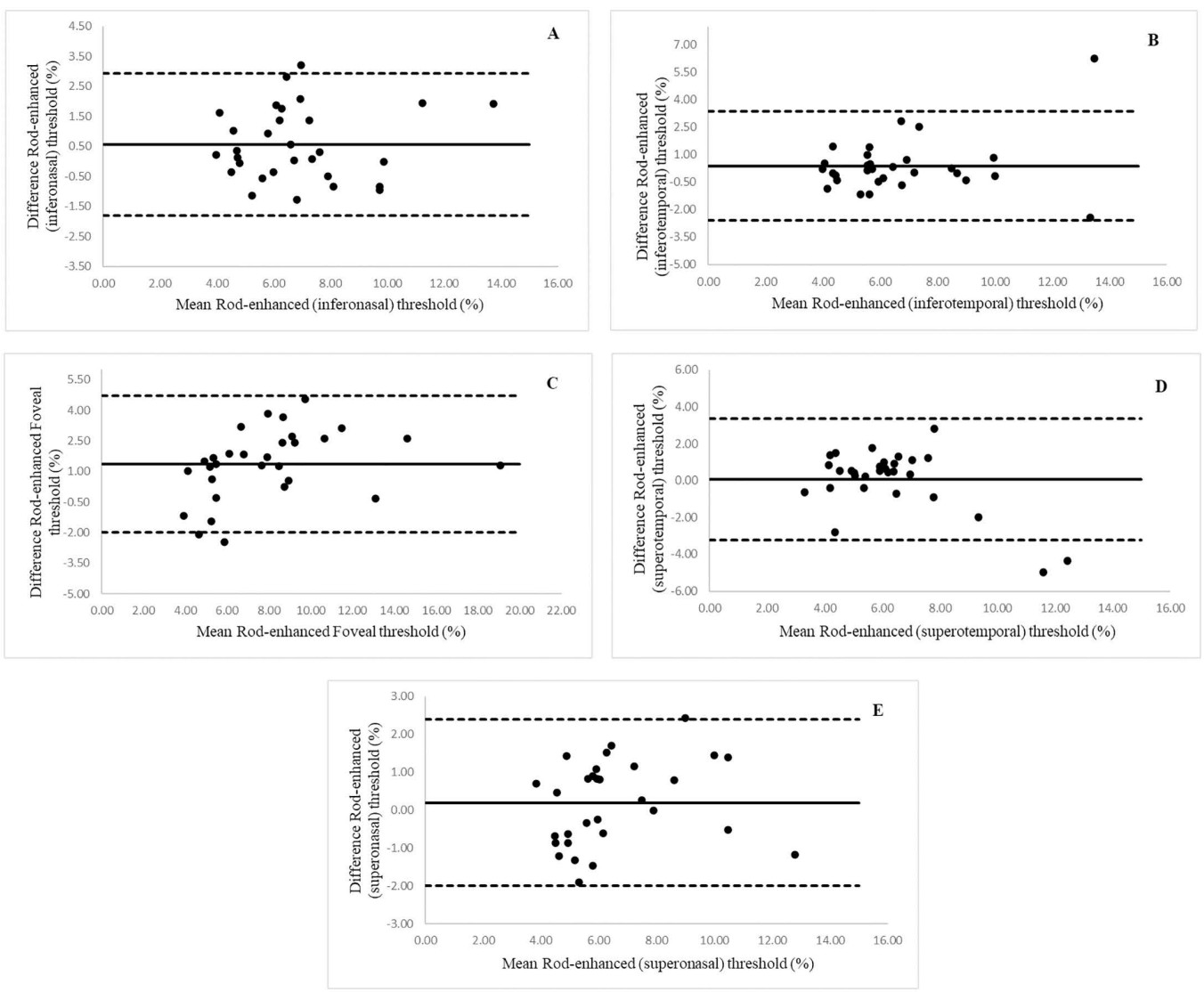

**Fig 3. Bland Altman plots to show between session repeatability.** (A) inter-visit agreement of inferonasal (−135) rod-enhanced FMT, (B) inter-visit agreement of inferotemporal (−45) rod-enhanced FMT, (C) inter-visit agreement of foveal (0) rod-enhanced FMT, (D) inter-visit agreement of superotemporal (45) rod-enhanced FMT, (E) inter-visit agreement of superonasal (135) rod-enhanced FMT. Middle horizontal solid line represents mean difference between visits 1 and 2 for rod-enhanced FMT. Upper and lower LoA (mean±1.96) represented by dashed line.

spread as compared to peripheral locations. Mean bias, upper and lower 95% LoA, and CoR values are given in Tables 2 and 3, respectively.

When sub analysis was carried out for different age groups, the CoR was found to be higher in the older age group for all conditions and locations (Table 4). It can also be seen that thresholds were higher for all locations in those people (n = 15) aged 45 years and over, compared to younger individuals (n = 15) ($p < 0.05$, independent sample t-test). In part, the effect of age on CoR is likely to reflect the higher thresholds expressed in older adults. This was further investigated by expressing the CoR as a proportion of the mean FMT for each condition for each age group. It can be seen that the

**Table 4. Age related CoR values for cone and rod mediated responses for 5 testing locations.**

| Tested Measures | Subjects <45 years | | | | Subjects >45 years | | | |
|---|---|---|---|---|---|---|---|---|
| | Mean FMT | SD | CoR | CoR% | Mean FMT | SD | CoR | CoR% |
| Cone Inferonasal (−135) | 4.81 | 1.04 | 1.72 | 35.80 | 6.00 | 1.76 | 1.78 | 29.67 |
| Cone Inferotemporal (−45) | 4.39 | 1.00 | 1.28 | 29.17 | 5.94 | 1.54 | 2.85 | 51.38 |
| Cone Foveal (0) | 4.11 | 1.43 | 2.17 | 52.75 | 6.04 | 2.39 | 4.06 | 67.15 |
| Cone Superotemporal (45) | 4.19 | 0.83 | 1.41 | 33.77 | 5.42 | 1.72 | 2.03 | 37.45 |
| Cone Superonasal (135) | 4.18 | 0.84 | 1.69 | 40.45 | 5.18 | 1.81 | 2.31 | 44.67 |
| Rod Inferonasal (−135) | 5.62 | 1.14 | 1.90 | 33.87 | 8.08 | 2.39 | 2.76 | 34.14 |
| Rod Inferotemporal (−45) | 5.25 | 1.12 | 1.76 | 33.46 | 8.13 | 2.64 | 3.88 | 47.68 |
| Rod Foveal (0) | 6.01 | 1.60 | 3.14 | 52.31 | 9.92 | 3.55 | 3.11 | 31.39 |
| Rod Superotemporal (45) | 5.03 | 1.02 | 2.04 | 40.64 | 7.45 | 2.14 | 4.21 | 56.51 |
| Rod Superonasal (135) | 5.43 | 1.11 | 1.93 | 35.45 | 7.67 | 2.38 | 2.38 | 31.07 |

repeatability was more consistent between age groups when expressed in proportional terms but was still poorer for the older participants for 7/10 locations.

The correlation and agreement between the two FMT measures was also estimated by ICC analysis (Table 5). Based on the ICC results, the test-retest reliability of Flicker-*Plus* test is good (0.75–0.9) to excellent (>0.9) for all measures [28]. ICC was similar in all locations and for cone and rod-enhanced conditions.

The three participants who took part in the intra-subject variability analysis were aged 29, 30, and 36 years (subjects 1, 2 and 3, respectively). Intra subject variability of Flicker-*Plus* test was assessed for both rod and cone-enhanced stimulus conditions, for each of the three subjects repeating the test 16 times, binocularly. Four test runs were performed per visit. Tables 6 and 7 show the mean cone and rod-enhanced FMT for each of the 5 locations repeated 16 times, SD, and the coefficient of variability (CoV) values respectively. The results showed that the foveal cone-enhanced FMT had a variability of 18.82%, 26.28% and 25.04%, whilst the results for rod-enhanced conditions had a variability of 29.29%, 26.01% and 22.48% for subjects 1, 2 and 3 respectively. Figs 4A and 4B shows the foveal cone and rod mediated FMTs for three subjects, in cone and rod enhanced conditions, respectively. Four test runs were carried out per visit. For participants 1 and 2, there was a tendency towards reducing thresholds over the visits, with the values leveling out after visit 2 (8 runs). This finding suggests that, for some individuals, at least 2 visits with multiple iterations are required to eliminate the learning effect. A similar trend was observed in the peripheral locations for both cone and rod mediated FMTs.

**Table 5. ICC estimates with 95% confidence interval for cone and rod-enhanced FMT.**

| Tested Measures | ICC | 95% confidence interval | |
|---|---|---|---|
| | | Lower bound | Upper bound |
| Cone Inferonasal (−135) | 0.90 | 0.77 | 0.96 |
| Cone Inferotemporal (−45) | 0.83 | 0.62 | 0.92 |
| Cone Foveal (0) | 0.84 | 0.67 | 0.93 |
| Cone Superotemporal (45) | 0.87 | 0.66 | 0.95 |
| Cone Superonasal (135) | 0.86 | 0.67 | 0.94 |
| Rod Inferonasal (−135) | 0.91 | 0.79 | 0.96 |
| Rod Inferotemporal (−45) | 0.90 | 0.79 | 0.95 |
| Rod Foveal (0) | 0.90 | 0.59 | 0.96 |
| Rod Superotemporal (45) | 0.84 | 0.65 | 0.92 |
| Rod Superonasal (135) | 0.93 | 0.86 | 0.97 |

**Table 6. Showing mean cone-enhanced flicker modulation thresholds for 5 locations tested by flicker-plus test, SD, and the CoV (%) for each subject.**

| Meridian | Subject 1 | | | Subject 2 | | | Subject 3 | | |
|---|---|---|---|---|---|---|---|---|---|
| | Mean | SD | CoV(%) | Mean | SD | CoV(%) | Mean | SD | CoV(%) |
| −135 | 2.25 | 0.33 | 14.62 | 1.83 | 0.35 | 19.11 | 3.22 | 0.51 | 15.69 |
| −45 | 2.38 | 0.31 | 13.06 | 1.98 | 0.30 | 15.42 | 3.29 | 0.47 | 14.47 |
| 0 | 1.81 | 0.34 | 18.82 | 2.53 | 0.66 | 26.28 | 3.47 | 0.86 | 25.04 |
| 45 | 2.05 | 0.27 | 13.44 | 1.92 | 0.39 | 20.4 | 2.53 | 0.41 | 16.04 |
| 135 | 1.96 | 0.33 | 17.06 | 1.98 | 0.37 | 18.89 | 2.51 | 0.53 | 21.24 |

**Table 7. Showing mean rod-enhanced flicker modulation thresholds for 5 locations tested by flicker-plus test, SD, and the CoV (%) for each subject.**

| Meridian | Subject 1 | | | Subject 2 | | | Subject 3 | | |
|---|---|---|---|---|---|---|---|---|---|
| | Mean | SD | CoV(%) | Mean | SD | CoV(%) | Mean | SD | CoV(%) |
| −135 | 2.12 | 0.29 | 13.79 | 2.28 | 0.30 | 13.44 | 2.59 | 0.26 | 9.99 |
| −45 | 1.87 | 0.24 | 13.26 | 1.99 | 0.37 | 18.79 | 2.12 | 0.36 | 16.94 |
| 0 | 1.89 | 0.55 | 29.29 | 3.14 | 0.81 | 26.01 | 2.79 | 0.63 | 22.48 |
| 45 | 1.86 | 0.28 | 15.28 | 2.00 | 0.44 | 22.04 | 1.98 | 0.32 | 15.96 |
| 135 | 1.86 | 0.29 | 15.49 | 2.19 | 0.28 | 12.79 | 2.25 | 0.27 | 12.19 |

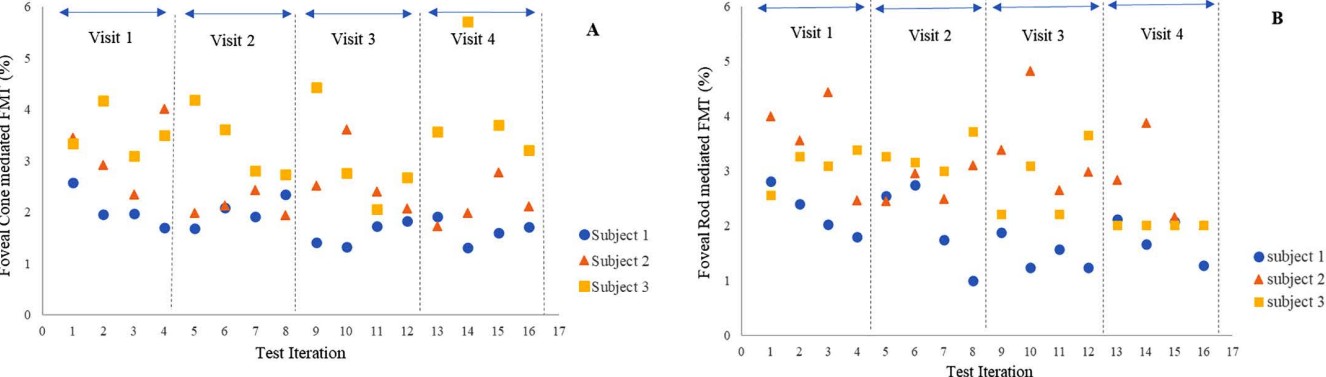

**Fig 4. Scatter plot showing foveal cone and rod mediated FMTs in subject 1, 2 and 3.** (A) Foveal cone mediated FMTs in each subject. (B) Foveal rod mediated FMTs in each subject.

## Discussion

The Flicker-*Plus* test was used to examine the repeatability of flicker modulation sensitivity (FMS) to enhance its potential value in clinical and research settings. This is the first paper demonstrating the repeatability of the Flicker-*Plus* test for assessing FMTs in healthy individuals across two sessions. The ICC values obtained for the between session repeatability suggested 'good' to 'excellent' agreement between visits [28]. This was the case at all locations, under both cone and rod-enhanced conditions. However, from a practical viewpoint, a researcher employing this test needs to have a quantifiable measure of the expected variation in the parameters between visits in order to appropriately power study design,

and to determine what magnitude of difference constitutes a clinically significant change [27,30]. To enable this, analysis based on the techniques described by Bland and Altman was conducted [26].

According to the Bland and Altman graphical analysis, a bias was seen in all measurements, with visit 2 showing a lower mean threshold for all conditions. This indicates the possible existence of a learning effect. Although the paired t-test showed that this difference was not statistically significant in this sample, the sample size of 30 is only powered to detect a medium effect size of 0.5 so a smaller effect may be present [31]. This is of importance in studies monitoring flicker thresholds over time, as sufficient training will be required to ensure that any perceived changes are not due to the learning effect alone. Multiple measurements were taken from three individuals to explore this further. These data showed a decrease in threshold over subsequent visits in two of the three participants, which only leveled out after four repeats over two visits. However, thresholds tended to increase again after multiple repeats in one visit, suggesting that fatigue effects may become significant upon prolonged repeated testing. It should also be noted that the participants who took part in the intra-subject variability test were familiar with psychophysical testing. The time course of diminution of learning effects may be different for non-trained observers. The learning effect is considered an important artefact in several visual psychophysical tests [32]. Previous literature investigating the learning effect between examinations using standard automated perimetry (SAP) shows an improvement in thresholds, with increasing familiarity of the test in both normal and glaucoma subjects [30,32,33]. Yenice and Temel (2005) reported a statistically significant difference in visual field indices (mean deviation and pattern standard deviation) on the 2nd visit (repeated within a week), due to a learning effect. Some studies report a minimal learning effect in patients with glaucoma and previous experience with perimetry, with more prominent learning effect between first and second examination and a persistent but stable effect between third and fourth visits [32]. However, there is significant variation between individuals, and individualised interpretation of results should be practiced in clinical settings. The Bland and Altman plots demonstrate a certain level of variability between visits, as evidenced by the spread of the LoA around the mean.

Characterising the CoR is a crucial statistical tool in study design. Whether or not a test is considered to be 'sufficiently' repeatable depends on the expected magnitude of the disease effect, as a change over time exceeding the CoR may be considered to be a clinically significant decline in visual performance. For example, according to the CoR values obtained, a change in foveal cone enhanced FMT exceeding 3.25 (or ~64%) would be considered to exceed the test-retest variability, and so would be considered significant. This means that in a cohort study design it would be inappropriate to power the study to detect a mean change over time smaller than the test-retest CoR [26]. The CoR for the more peripheral locations was lower, suggesting that the test is able to detect a smaller magnitude of change with the stimulus presented at peripheral locations. For example, for the cone-enhanced test in the superotemporal location, the CoR was 1.81, indicating that a change exceeding 1.81 (38%) in FMT would be clinically significant. For the rod enhanced test, a similar pattern was noted. Whilst the absolute repeatability of rod FMTs tended to be poorer than for cones, the CoR were more comparable when expressed as a percentage of the mean FMT. For example, the % CoR for rod FMT was 34.6% at the inferonasal location, as compared to 32.2% for the same location for the cone enhanced condition.

It is also important to consider that CoR may not be a consistent value for all populations. In the sub analysis conducted we demonstrated that CoR was elevated in people ≥45 years of age. FMT itself has also been shown to increase with increased age [25], so we also evaluated CoR as a proportion of mean FMT for each sub group (see Table 4). The % CoR remained higher in the older age group for 7/10 conditions. This is consistent with other literature in different aspects of psychophysical testing [1,25]. It is also likely that the CoR would be further elevated in people with pathology [34]. It has been shown, for example, in SAP that the variability was higher with visual field loss in glaucoma patients [35,36]. Hence, it would be of value for researchers to carry out further work to determine the CoR in their population group of interest. The study conducted by McKeague, et al., (2014), reported a CoR of 53.4% of the mean threshold of 14 Hz flicker test. Our study demonstrated comparable repeatability, with mean % CoR across locations of 44.1% and 41.8% for cone and rod enhanced stimuli, respectively.

Whilst there is no evidence to date on the expected magnitude of elevation of FMT in people with different patholo-gies, it is possible to examine other flicker detection studies to determine what change might be expected in proportional terms. Temporal contrast sensitivity (TCS) thresholds are reported to be higher in the people with glaucoma as compared to controls [17,18,20,34,37–42]. Although the number of studies investigating TCS in glaucoma are limited, there is still some evidence that this deficit may occur prior to perimetric defects [18], and of an increasing decline in flicker sensitiv-ity with increasing disease severity [17]. Fidalgo et al., (2018) tested peripheral photopic flicker thresholds in glaucoma patients with varying disease severity, showing 123% increase in thresholds as disease progress from early to moderate stage, and 58% increase between moderate and severe disease stages, which is greater than the % CoR for peripheral cone-enhanced condition (32%−44%). Flicker sensitivity is also reduced in individuals with early stages of AMD, and it decreases further with the disease severity [9,10,15]. Another study reported the mean flicker thresholds measured using Flicker-*Plus* test in individuals with neovascular AMD (n = 13) were significantly higher (262%), which significantly exceeds the % CoR of 43.4% reported here for a comparable test condition [43]. Luu et al., (2013) assessed flicker sensitivity using flicker perimetry in AMD patients and reported the test-retest reproducibility with fixed bias of −0.41dB and 95% LoA were ± 5dB, suggesting that smaller changes should be interpreted carefully. The FMT for foveal rod-enhanced condition showed a fixed bias of 1.36 with LoAs −1.98 and 4.70.

## Conclusion

The findings show good to excellent test-retest repeatability of Flicker-*Plus* test. The CoR was higher (indicating poorer repeatability) for rod-enhanced conditions and at the fovea. There are many factors that can cause this increased variabil-ity, and these will be examined in future studies.

Decrease in temporal contrast sensitivity occur as the number of photoreceptors diminishes [10]. Most of the flicker tests have focused on foveal location which is more populated with cone photoreceptors hence assessing the cone medi-ated flicker sensitivity. It is also reported that the number of cones remains relatively constant throughout human life span [[44]] and thus the effect of age and disease cannot be separated very precisely using these tests. There is evidence that temporal sensitivity assessed at peripheral locations would provide a better measure of visual function loss in diseases such as AMD and glaucoma, in which early changes are not confined to central retina [11]. The excellent repeatability of the rod and cone FMT assessed in the parafoveal retinal locations suggest that this test would be valuable in assessing this functional loss. Sub-analysis also indicated that CoR was higher in older adults (>45 years), both in absolute terms and when expressed as a proportion of the mean FMT. This should be considered in sample size calculations for clinical trial design.

This study will assist clinicians in determining whether reported changes over time are caused by measurement inaccu-racy or disease advancement, assuming consistent experimental setting and psychophysical techniques.

## Acknowledgments

We acknowledge the J&P Wolf Trust for supporting AH and SB with bursaries for their doctoral studies. No other funding or sponsorship was received to support this study or publication of this article. All named authors meet the International Committee of Medical Journal Editors (ICMJE) criteria for authorship for this article, take responsibility for the integrity of the work as a whole, and have given their approval for this version to be published.

## Author contributions

**Conceptualization:** Aiman Hafeez, Alison Binns, Sajni Bohra, Irene Ctori, John L. Barbur.

**Data curation:** Aiman Hafeez, Sajni Bohra.

**Investigation:** Aiman Hafeez, Sajni Bohra.

**Methodology:** Aiman Hafeez, Alison Binns, Sajni Bohra, Irene Ctori, John L. Barbur.

**Validation:** Aiman Hafeez, Alison Binns, Sajni Bohra, Irene Ctori, John L. Barbur.

**Visualization:** Aiman Hafeez, Alison Binns, Sajni Bohra, Irene Ctori, John L. Barbur.

**Writing – original draft:** Aiman Hafeez.

**Writing – review & editing:** Aiman Hafeez, Alison Binns, Irene Ctori, John L. Barbur.

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
