## [Decision Letter · Decision Letter 0]

21 Jul 2025

Repeatability of flicker modulation sensitivity measured using the Flicker-Plus test

PLOS ONE

Dear Dr. Hafeez,

Thank you for submitting your manuscript to PLOS ONE. After careful consideration, we feel that it has merit but does not fully meet PLOS ONE’s publication criteria as it currently stands. Therefore, we invite you to submit a revised version of the manuscript that addresses the points raised during the review process.

The manuscript was well-received by the reviewers. Nevertheless, some minor changes are required.

https://journals.plos.org/plosone/s/submission-guidelines#loc-laboratory-protocols . Additionally, PLOS ONE offers an option for publishing peer-reviewed Lab Protocol articles, which describe protocols hosted on protocols.io. Read more information on sharing protocols at https://plos.org/protocols?utm_medium=editorial-email&utm_source=authorletters&utm_campaign=protocols .

We look forward to receiving your revised manuscript.

Kind regards,

Giulio Contemori, Ph.D.

Academic Editor

PLOS ONE

Journal Requirements:

[I have read the journal's policy and the authors of this manuscript have the following competing interests: Author AB has a consultancy agreement with Boehringer Ingelheim. The authors (AH, SB, JLB, IC) report no competing interest and have no proprietary interest in any of the materials mentioned in this article.]. 

We note that you potentially received funding from a commercial source: [Boehringer Ingelheim]

Additional Editor Comments:

The manuscript was well-received by the reviewers. Nevertheless, some changes are required.

Reviewers' comments:

Reviewer's Responses to Questions

**Comments to the Author**

1. Is the manuscript technically sound, and do the data support the conclusions?

Reviewer #1: Yes

Reviewer #2: Yes

2. Has the statistical analysis been performed appropriately and rigorously?

Reviewer #1: Yes

Reviewer #2: Yes

3. Have the authors made all data underlying the findings in their manuscript fully available?

Reviewer #1: Yes

Reviewer #2: Yes

4. Is the manuscript presented in an intelligible fashion and written in standard English?

Reviewer #1: Yes

Reviewer #2: Yes

Reviewer #1: This is an important study evaluating test-retest reproducibility of flicker modulation thresholds recorded in normal subjects under cone and rod enhancing conditions. The results are sound and well presented. I would only ask the authors to report the units (linear, log) for flicker modulation thresholds in their plots.

Reviewer #2: The test is useful and an important study to carry out in understanding the COR in flicker thresholds. Hafeez et al reported inter-session repeatability of cone and rod-enhanced flicker thresholds.

I have a few minor points

1. Is there any reference or justification for testing 16 times in a small subgroup of individuals to assess intraobserver variability? Why not 10 times for example?

2. How did the examiner ensure that participant did not move their eyes during the test? Do they have an equivalent of eye tracking such as Humphreys visual field. If not? How do you justify the parafoveal measurements are truly parafoveal measurements?

3. Is there an acceptable “COR” that is used in Ophthalmic instrumentation based on literature review that the authors can compare theirs against?

4. The exact age distribution of individuals is not clear.

5. It would be worth also plotting FMT thresholds against age as one additional figure to see age-related changes, if any.

**Do you want your identity to be public for this peer review?** For information about this choice, including consent withdrawal, please see our Privacy Policy

Reviewer #1: **Yes: ** Benedetto Falsini

Reviewer #2: No

---

## [Author Response · Author response to Decision Letter 1]

21 Aug 2025

Dear Dr Contemori,

We are pleased to submit the revised version of our manuscript titled "Repeatability of flicker modulation sensitivity measured using the Flicker-Plus test" (Manuscript ID: PONE-D-25-20564) for consideration for publication in PLoS ONE.

We sincerely thank you and the reviewers for their constructive feedback and valuable suggestions, which have significantly improved the quality of our manuscript. We have carefully addressed all the comments and concerns raised by the reviewers and have made substantial revisions accordingly. Some minor additional changes have been made to improve clarity.

Below, we provide a detailed point-by-point response to each reviewer's comments, indicating how we have addressed each concern and the corresponding changes made in the revised manuscript. All changes in the manuscript are highlighted using track changes/different coloured text for easy identification.

We believe that the revised manuscript now addresses all the concerns raised and provides a more comprehensive and robust study. We hope that you will find the revisions satisfactory and that the manuscript is now suitable for publication in PLoS ONE.

Thank you for your time and consideration. We look forward to your favourable response.

Please do not hesitate to contact me should you require any additional information.

Best regards,

Aiman Hafeez

School of Health and Medical Sciences

Department of Optometry and Visual Science

City, St George’s University of London, EC1V 0HB

London, United Kingdom

aiman.hafeez@citystgeorges.ac.uk

Detailed response to reviewers

Reviewer #1:

This is an important study evaluating test-retest reproducibility of flicker modulation thresholds recorded in normal subjects under cone and rod enhancing conditions. The results are sound and well presented. I would only ask the authors to report the units (linear, log) for flicker modulation thresholds in their plots.

Response: As suggested the units have been added to the plots (fig. 2A-2E, and 3A-3E) and uploaded seperately. The flicker modulation thresholds are expressed on a linear scale as a percentage of the background luminance. This has been added in the manuscript body as well (Page 7, Line: 183-184).

Reviewer #2:

1. Is there any reference or justification for testing 16 times in a small subgroup of individuals to assess intraobserver variability? Why not 10 times for example?

Response: Thank you for this question. The statistical distribution of sample means of 10 or 16 measurements is normal and follows the central limit theorem, so indeed we could have chosen to average only 10 samples. Increasing the number of measurements reduces the standard error and improves the reliability of the estimate, so the larger the sample size the more accurate the estimate of the standard deviation associated with single measurements. We selected 16 measurements simple because of practical limitations such as the time and effort needed to complete the tests. Our aim was to balance statistical robustness with experimental practicality in our investigation. These factors led to the conclusion that 16 repeats were a suitable compromise.

2. How did the examiner ensure that participant did not move their eyes during the test? Do they have an equivalent of eye tracking such as Humphreys visual field. If not? How do you justify the parafoveal measurements are truly parafoveal measurements?

Response: Although formal eye-tracking equipment (such as that used in Humphrey visual field testing) was not employed in this study, measures were taken to ensure that participants maintained central fixation and that parafoveal measurements were as reliable as possible.

Throughout the test, participants received specific instructions to focus on the centre of their visual field. The participants were verbally encouraged by the examiner throughout the test session. The stimulus arrangement itself was carefully designed to facilitate fixation: a brief presentation of a fixation target (a central cross enclosed in a square frame) is presented briefly before each stimulus. Peripheral guides pointing towards the centre of the screen (as shown in Fig. 1) also helped stabilise the participant's point of regard.

Furthermore, there was little time for exploratory eye movements or fixation shifts during stimulus exposure due to the brief presentation of the visual stimuli. Even though we recognise that involuntary eye movements cannot be totally ruled out in the absence of eye-tracking. Spatially structured fixation stimuli, brief stimulus presentation time, and positive regular feedback all contributed to a reduced likelihood of eye movements for parafoveal testing.

3. Is there an acceptable “COR” that is used in Ophthalmic instrumentation based on literature review that the authors can compare theirs against?

Response: In the context of ophthalmic instrumentation and psychophysical testing (like flicker perimetry or flicker threshold assessments), the Coefficient of Repeatability (COR) is a commonly used metric to evaluate test-retest reliability. However, there is no single universally accepted COR threshold across all ophthalmic tests. It often depends on the specific instrument or test type (e.g., standard automated perimetry, flicker perimetry, photopic vs scotopic conditions), the population tested (e.g., healthy subjects vs glaucoma patients) and the measurement scale (e.g., log units, decibels, percentage contrast). While there is no single fixed “COR,” published but the literature provides normative data and age-related reference ranges are commonly accepted for comparing flicker threshold measurements in ophthalmic instrumentation.

The study “An Evaluation of Two Candidate Functional Biomarkers for AMD” conducted by McKeague et al., (2014), reported a COR of 53.4% of the mean threshold of 14Hz flicker test. Our study demonstrated comparable repeatability, with mean % CoR across locations of 44.1% and 41.8% for cone and rod enhanced stimuli, respectively. Our findings revealed location-dependent variability, with foveal locations exhibiting higher % CoR than parafoveal locations, particularly for cone stimuli. This spatial variation in repeatability provides additional insight into the test's performance characteristics and suggests that measurement precision varies systematically across retinal locations, which is consistent with the known spatial heterogeneity of retinal function.

4. The exact age distribution of individuals is not clear.

Response: Table 1 is added in results section showing the age distribution of all participants. (Page 8, Line: 227-229). The corresponding table numbers are changed and also in the manuscript body.

Table 1. Total number of participants recruited and participants per decade with mean age (M) and standard deviation (SD).

  Number of Participants (n)  Mean age (M ± SD) 

Total 30  44 ± 15.00

18 - 29 6  25 ± 4.32

30 - 39 6 33 ± 3.38

40 - 49 6  44 ± 1.21

50 - 59 6 53 ± 2.58

60 - 69 5  66 ± 1.30

≥ 70 1 71 ± 0.00

5. It would be worth also plotting FMT thresholds against age as one additional figure to see age-related changes, if any.

Response: We agree that it is interesting to investigate the relationship between flicker modulation thresholds (FMT) and age. To evaluate age-related effects, however, the current study was not specifically powered or designed or designed to achieve this aim. Given the constraints on the data set, we believed that an analysis of such a nature would run with the potential of being overinterpreted, and the sample size and age distribution were insufficient to make reasonable conclusions regarding age-related trends.

We recognise the significance of this observation, and a future study with a larger and more age-diverse population will be conducted to examine the impact of ageing on FMTs.

Academic Editor's comments

Response: The corresponding author has double checked the style requirements for PLOS ONE and made changes to the file name, table and figure title caption.

2. Thank you for stating the following in the Competing Interests section: We note that you potentially received funding from a commercial source: [Boehringer Ingelheim]. Please include your amended Competing Interests Statement within your cover letter. We will change the online submission form on your behalf.

Response:

Competing Interests Statement:

Author AB has a consultancy agreement with Boehringer Ingelheim. However, Boehringer Ingelheim did not provide funding for this study and had no role in its design, execution, or interpretation. The authors (AH, SB, JLB, IC) declare that they have no competing interests and no proprietary interest in any of the materials mentioned in this article. This does not alter our adherence to PLOS ONE policies on sharing data and materials.

3. Please review your reference list to ensure that it is complete and correct. If you have cited papers that have been retracted, please include the rationale for doing so in the manuscript text or remove these references and replace them with relevant current references. Any changes to the reference list should be mentioned in the rebuttal letter that accompanies your revised manuscript. If you need to cite a retracted article, indicate the article’s retracted status in the References list and also include a citation and full reference for the retraction notice.

Response: The reference list has been reviewed and the only reference retracted has been removed and another reference is provided to support that statement “Aspinall PA. Some methodological problems in testing visual function. Mod Probl Ophthalmol. 1974;13(Journal Article):2–7.” (Page 14, line 338).

4. We note that you have indicated that there are restrictions to data sharing for this study. For studies involving human research participant data or other sensitive data, we encourage authors to share de-identified or anonymized data. However, when data cannot be publicly shared for ethical reasons, we allow authors to make their data sets available upon request.

b) If there are no restrictions, please upload the minimal anonymized data set necessary to replicate your study findings to a stable, public repository and provide us with the relevant URLs, DOIs, or accession numbers. “Please update your Data Availability statement in the submission form accordingly."

Response: As requested, the data availability statement is updated in the form. The minimal anonymized data set necessary to replicate our study findings is uploaded to Figshare repository and the relevant URLs, DOIs are given below.

“The dataset file will be available from Figshare Repository.

After discussion with the authors of the manuscript the data is deposited in the Figshare repository. The DOI is included in the statement which becomes active when the manuscript is published. A private URL (https://figshare.com/s/08bb3523e63d4294f313) is generated for PLOS ONE’s editors and reviewers for access.

Additional Improvements Made

Beyond addressing the reviewers' comments, we have also made the following additional improvements:

• Some minor additional changes (grammar/language improvements) have been made to improve clarity. (Line 20, 35, 39, 41, 49-50, 53-54, 61,76, 87, 111, 178-180, 243, 292, 338, 358-359, 375-377, 395, 400-402)

• In acknowledgments section we have added the funding received by two authors AH and SB who are doctoral researcher and received bursaries from J&P Wolf Trust. This funding did not have any role in this study’s design, execution, or interpretation.

Compliance Statements

We confirm that all authors have approved the revised manuscript. The study complies with PLoS ONE's ethical guidelines. All data supporting the conclusions are available within the manuscript.

Thank you once again for the opportunity to revise and resubmit our manuscript.

---

## [Decision Letter · Decision Letter 1]

8 Sep 2025

Repeatability of flicker modulation sensitivity measured using the Flicker-Plus test

PONE-D-25-20564R1

Dear Dr. Hafeez,

We’re pleased to inform you that your manuscript has been judged scientifically suitable for publication and will be formally accepted for publication once it meets all outstanding technical requirements.

Kind regards,

Giulio Contemori, Ph.D.

Academic Editor

PLOS ONE

Additional Editor Comments (optional):

Reviewer #1:

Reviewer #2:

Reviewers' comments:

Reviewer's Responses to Questions

**Comments to the Author**

Reviewer #1: All comments have been addressed

Reviewer #2: All comments have been addressed

2. Is the manuscript technically sound, and do the data support the conclusions?

Reviewer #1: (No Response)

Reviewer #2: Yes

3. Has the statistical analysis been performed appropriately and rigorously?

Reviewer #1: (No Response)

Reviewer #2: Yes

4. Have the authors made all data underlying the findings in their manuscript fully available?

Reviewer #1: (No Response)

Reviewer #2: Yes

5. Is the manuscript presented in an intelligible fashion and written in standard English?

Reviewer #1: (No Response)

Reviewer #2: Yes

Reviewer #1: (No Response)

Reviewer #2: All the comments have been addressed satisfactorily. I have no further comments to add on to the manuscript.

**Do you want your identity to be public for this peer review?** For information about this choice, including consent withdrawal, please see our Privacy Policy

Reviewer #1: **Yes: ** Benedetto Falsini

Reviewer #2: No

---

## [Editor Report · Acceptance letter]

PONE-D-25-20564R1

PLOS ONE

Dear Dr. Hafeez,

I'm pleased to inform you that your manuscript has been deemed suitable for publication in PLOS ONE. Congratulations! Your manuscript is now being handed over to our production team.

Kind regards,

on behalf of

Dr. Giulio Contemori

Academic Editor

PLOS ONE